# Preprocessing Selection for Deep Learning Classification of Arrhythmia Using ECG Time-Frequency Representations

Rafael Holanda *[ID], Rodrigo Monteiro *[ID] and Carmelo Bastos-Filho *[ID]

Department of Computer Engineering, Polytechnic School, University of Pernambuco,
Recife 50720-001, PE, Brazil
* rbbh@ecomp.poli.br (R.H.); rodrigo.paula@unicap.br (R.M.); carmelofilho@ieee.org (C.B.-F.)

**Abstract:** The trend of using deep learning techniques to classify arbitrary tasks has grown significantly in the last decade. Such techniques in the background provide a stack of non-linear functions to solve tasks that cannot be solved in a linear manner. Naturally, deep learning models can always solve almost any problem with the right amount of functional parameters. However, with the right set of preprocessing techniques, these models might become much more accessible by negating the need for a large set of model parameters and the concomitant computational costs that accompany the need for many parameters. This paper studies the effects of such preprocessing techniques, and is focused, more specifically, on the resulting learning representations, so as to classify the arrhythmia task provided by the ECG MIT-BIH signal dataset. The types of noise we filter out from such signals are the Baseline Wander (BW) and the Powerline Interference (PLI). The learning representations we use as input to a Convolutional Neural Network (CNN) model are the spectrograms extracted by the Short-time Fourier Transform (STFT) and the scalograms extracted by the Continuous Wavelet Transform (CWT). These features are extracted using different parameter values, such as the window size of the Fourier Transform and the number of scales from the mother wavelet. We highlight that the noise with the most significant influence on a CNN's classification performance is the BW noise. The most accurate classification performance was achieved using the 64 wavelet scales scalogram with the Mexican Hat and with only the BW noise suppressed. The deployed CNN has less than 90k parameters and achieved an average F1-Score of 90.11%.

**Keywords:** arrhythmia; classification; time-frequency; preprocessing; deep learning

## 1. Introduction

According to the World Health Organization (WHO) [1], and referenced in studies [2], chronic cardiovascular diseases (CVDs) are considered to be one of the leading causes of mortality worldwide, especially when considering people dealing with problems concerning alcohol consumption. A sub-field of CVDs is arrhythmia, which are heartbeats that do not follow the pace of regular beats, causing the heart to beat either faster or slower, depending on the type of arrhythmia [3,4].

Multiple works in the literature attempt to accurately classify these diseases, using machine learning, deep learning models, and signal processing techniques. Electrocardiogram (ECG) signals are treated by eliminating noise, extracting essential features, and, thus, facilitating information representation within the models.

The fact that works in the state-of-the-art acquire very high classification metrics is unquestionable. However, many of these works use heavy deep-learning models with many parameters, making such solutions impracticable for embedding in devices with limited storage and battery resources.

Another gap noticed by the authors of the present work is that most of the literature studies have not displayed a parameter search methodology for the signal processing pipeline in their works. The goal is to achieve the best combination of preprocessing

parameters and techniques, their resulting learning representations, and how they affect the classification performance of model.

These issues are relevant because a lower computational cost may be more important than a high classification metric, depending on a project's classification and optimization requirements. Learning representations resulting from the right set of preprocessing techniques may aid the task of model in classifying examples more efficiently, reducing model complexity while not losing much in classification performance.

The work of Zhang et al. [5], for instance, used a combination of the Wigner–Ville distribution (WVD) [6] and Hilbert transforms (HTs) [7] to generate time-frequency representations and input them to a pre-trained CNN in order to classify arrhythmias. This work achieved an overall F1-Score of 0.9595. However, the model used was a Convolutional Neural Network (CNN) with over 44 million parameters (Resnet 101 [8]), and they also never discussed the parameters used to generate their time-frequency representations.

As for Guo et al. [9], they used only temporal features as input to an ensemble of a 1D CNN and a Gated Recurrent Network (GRU), requiring 18 million parameters to achieve an 89.75% F1-Score in one of the target classes.

Another work, [10], used the Continuous Wavelet Transform (CWT) as the time-frequency feature extraction method and acquired 99.41% sensitivity, 98.91% specificity, 99.39% positive predictive value, and 99.23% overall accuracy. Instead of varying the learning representation parameters, they fixed them and varied the CNN's hyperparameters and structural parameters (convolutional and max-pooling kernel size and the number of fully-connected neurons, among others). This methodology is not what we are looking for, as our goal is not, as of yet, to find the best model for a classification task, but rather to understand how the different learning representations may interfere with the learning procedure of a specific model for such a task.

The work of Qurraie and Afkhami [11] used the WVD as its time-frequency representation and fed it, along with other manually-extracted statistical features, to an ensemble of 100 decision trees combined in a bagging scheme. They achieved sensitivity and positive predictive values of 99.67% and 98.92%, respectively. The choice for this machine learning model may have an advantage over a deep learning model in terms of computational complexity. However, they did not provide a metric, such as the number of Floating Point Operations (FLOPs). Even though they explained the choice of window representations for the WVD, they did not provide any variation of the parameters, their effects on the representations, and their subsequent effects on the model's performance.

Wu et al. [12] compared the following three different time-frequency extraction methods, which they used as input to a low-complexity CNN: the Short-time Fourier Transform (STFT), CWT, and the Pseudo Wigner-Ville distribution (P-WVD). They listed all the parameters used to extract their specific learning representations, but they never showcased their selection methodology for such parameters.

The two gaps we wanted to fill, by means of this study, are the lack of an explainable parameter selection that results in the learning representations of ECG signals and the use of a low-complexity deep learning model for arrhythmia tasks. We propose an explainable and reproducible experimental methodology for the preprocessing of arrhythmia and the classification of its pipelines using a low computational cost model.

We built a search methodology on the parameters of two famous time-frequency feature extraction techniques, STFT and CWT. We also explored the use of filters in ECG signals and analyzed the effects of their presence or absence on learning representations and, consequently, on the model's classification performance. The model chosen for this work was a 2D CNN with under 90k parameters.

The sections in this work are organized as follows. In Section 2, we explain the experimental methodology used to solve the problem at hand. In Section 3, we briefly provide the experimental results and in Section 4 we discuss the results further. Finally, in Section 5 we conclude and share ideas for future work to enrich the present work.

## 2. Materials and Methods

### 2.1. Database

The database used for this work was the MIT-BIH Arrhythmia Database [13], which contains 48 excerpts of two-channel ambulatory ECG readings sampled at 360 samples per second and annotated by multiple specialists.

### 2.2. Preprocessing

The pipeline for the signal preprocessing is composed of the following steps:

1.  Beat segmentation;
2.  Beat labeling;
3.  Denoising (depending on the experimental setup);
4.  2D feature extraction

The signals went through each of these steps, in the order in which they are listed, before being fed into the model, which is explained in Section 2.3.

#### 2.2.1. Beat Segmentation

Regarding beat segmentation, two approaches, using concatenated beats or using single beats, have proved to work well in the literature. Each of the approaches has its own advantages and disadvantages.

The segmentation methodology chosen for this work was the latter, the single beats approach, whereby the beats were sampled by obtaining 100 samples before the R-peaks and 150 samples after, as performed in the works of Li et al. [14] and Zhang et al. [5]. This resulted in heartbeats with 250 samples each. We chose this type of segmentation, rather than the concatenated beats approach, so as to acquire more data and, thus, avoid using over-sampling strategies. The beats were segmented via code using the R-peak annotations provided by the MIT-BIH database.

#### 2.2.2. Beat Labeling

The MIT-BIH database is composed of beats in over ten classes. However, the American Association of Medical Instrumentation (AAMI) recommends classifying such beats into five subgroups: Normal Beat (N), Ventricular Ectopic Beat (VEB), Supraventricular Ectopic Beat (SVEB), Fusion Beat (F), and Unknown Beat (Q). This was the methodology chosen for this work. We decided to disregard class Q, which works like that conducted by Mondéjar et al. [15] have also done before, as this class is almost non-existent. This resulted in a total of 96,782 beats.

Table 1 shows the class division of the MIT-BIH database, along with the division recommended by the AAMI.

**Table 1.** MIT-BIH Class Label Division.

| MIT-BIH Heartbeat Class | AAMI Heartbeat Class |
|---|---|
| Normal (N)<br>Left Bundle Branch Block (L)<br>Right Bundle Branch Block (R) | Normal Beat (N) |
| Atrial premature beat (A)<br>Aberrated atrial premature beat (a)<br>Nodal (junctional) Premature (J)<br>Supraventricular Premature (S)<br>Atrial Escape (e)<br>Nodal Escape (j) | Supraventricular Ectopic Beat (SVEB) |
| Premature Ventricular Contraction (V)<br>Ventricular Escape (E) | Ventricular Ectopic Beat (VEB) |

**Table 1.** *Cont.*

| MIT-BIH Heartbeat Class | AAMI Heartbeat Class |
| --- | --- |
| Fusion of Ventricular and Normal (F) | Fusion Beat (F) |
| Paced (/)<br>Fusion of Paced and Normal (f)<br>Unclassified (Q) | Unknown Beat (Q) |

2.2.3. Denoising

The raw signals of the MIT-BIH database are contaminated by certain types of noise, which might result in deterioration of a model's classification performance and, therefore, may require denoising processing before being fed into a machine learning model.

A pervasive ECG noise is the Baseline Wander (BW) [16], a low-frequency noise, usually below 1 Hz, which is usually caused by electrodes due to the patient's breathing and movement. The method chosen to suppress this noise was a high pass second-order Butterworth filter with a cutoff frequency at 0.5 Hz, as employed in Wu et al. [12].

Another common noise type affecting the ECG is Powerline Interference (PLI) [16]. PLI is caused by electrical transmission lines affecting most electronic devices plugged into power sockets. It usually happens around the 50 Hz or 60 Hz frequencies. The method chosen to suppress this noise was a Notch filter, as performed by Sabut et al. [17].

2.2.4. Two-Dimensional Feature Extraction

The most common representation people think of is that of a signal being measured in the time spectrum. It is essential to acknowledge that there may be important information in the frequency spectrum of any signal.

An essential tool to visualize the frequency spectrum of an arbitrary time signal $x(t)$ is the calculation of the Fourier Transform [18]. The Fourier Transform represents signals in the frequency domain $\omega$ by computing an infinite sum of complex sinusoidal functions with different frequencies.

However, even with such a tool, one can only analyze information in the frequencies that compose such a signal, losing time domain information.

To work around this limitation, researchers have developed signal processing techniques that allow one to analyze the change of frequencies over time. These techniques allow information in both domains to be showcased and, eventually, mapped by machine learning and deep learning algorithms.

In the present work, two of these techniques are compared against each other: the Short-time Fourier Transform (STFT) [19] and the Continuous Wavelet Transform (CWT) [20].

The STFT is a parametric signal processing technique that applies the Fourier Transform along a signal $x(t)$ using a sliding window $g(t)$ to different parts of the signal,

$$X(\tau,\omega) = \int_{-\infty}^{+\infty} x(t) \cdot g(t - \tau) \cdot e^{-j\omega t} dt, \tag{1}$$

which allows one to analyze how the frequency varies over time by concatenating each of these frame measures.

The resulting feature $X(\tau,\omega)$ is a two-dimensional array called a spectrogram that showcases the variation of the frequencies $\omega$ in respect to the time frames $\tau$. The final representation of the spectrogram features

$$X(\tau,\omega)_{final} = \log_{10} |X(\tau,\omega)|^2, \tag{2}$$

was computed as the power spectrum given by a $\log_{10}$ normalization to approximate larger values to smaller ones, and, consequently, more uniformly distributing the energy across the features.

On the other hand, the CWT showcases the change of frequency over time in a fairly different manner than that of the STFT. Instead of sinusoidal functions, represented by $e^{-j\omega}$, it uses functions called wavelets, represented by $\psi(t)$. These functions are convolved on the reference signal $x(t)$ with different sets of scales $s$,

$$X(\tau,s) = \frac{1}{\sqrt{s}} \int_{-\infty}^{+\infty} x(t) \cdot \psi(\frac{t-\tau}{s}) dt, \tag{3}$$

and the feature $X(\tau,s)$ resulted from this operation, due to the nature of the variation of scales from the mother wavelet, is commonly called a scalogram.

The scales of the mother wavelet are inversely proportional to its center frequency. The relationship of the conversion from scale to frequency is given by

$$\omega(s) = \frac{\omega_c \cdot \omega_{sr}}{s}, \tag{4}$$

where $\omega_c$ is the mother wavelet's center frequency, $s$ is its scale and $\omega_{sr}$ is the sample rate.

Just as with the spectrograms, the final scalograms were computed with a $\log_{10}$ normalization on their power spectrum:

$$X(\tau,s)_{final} = \log_{10} |X(\tau,s)|^2. \tag{5}$$

Finally, to standardize the model's input and cut computational costs, the features were all resized to a dimension of $64 \times 64$ using bi-linear interpolation.

Some of these representations are displayed in Figure 1, along with their original time and frequency representations. The effect of the BW noise is showcased with a dashed red arrow on the first image from the top on the y-axis, as there was an offset from the zero-mean. In the second image, the effect of this same noise is showcased with a dashed red line around the peak of the 0 Hz frequency. The PLI noise, likewise, is showcased with a dashed magenta line around the 60 Hz peak. By looking at the frequency representation, it is clear that the noise with the most significant influence energy-wise was the BW noise. In the third image, the noises are indicated on the y-axis with the same colors. The high energy distribution across the spectrogram on the 0 Hz and 60 Hz frequencies can be seen.

In the first image from the bottom, the time representation with both noises suppressed is displayed, and the red square reference aids the visualization of the effect of suppressing the BW noise. The PLI noise suppression can also be noted if one looks closely at the signal, as there are fewer "spikes" along it. In the second image from the bottom, the same effects can be noted differently, with the cancellation of the noise frequencies caused by the high-pass Butterworth filter and the Notch filter. In the third image from the bottom, the energy distribution that corrupted the spectrogram disappeared from the 0 Hz and 60 Hz frequencies.

As for both scalograms, the interpretation of the effects of both noises on the representation is not so intuitive. However, one can visualize the different shape patterns on both scalograms, which display the effects of noise corruption and suppression on these features.

### 2.3. Model

The model used for the experiments was a low-complexity 2D Convolutional Neural Network (CNN). We considered it to be a low-complexity model because it has significantly fewer parameters than Resnet 101 [8], AlexNet [21], or VGG-16 [22].

The network had 12 layers, with only 5 having trainable parameters (the convolutional and dense layers) and a Softmax function at the end of the network to output probabilities and classify the samples. Only one layer was used for regularization, the dropout layer, with a 20% dropout rate. The model had 88,916 parameters, and its architecture is displayed in Table 2.

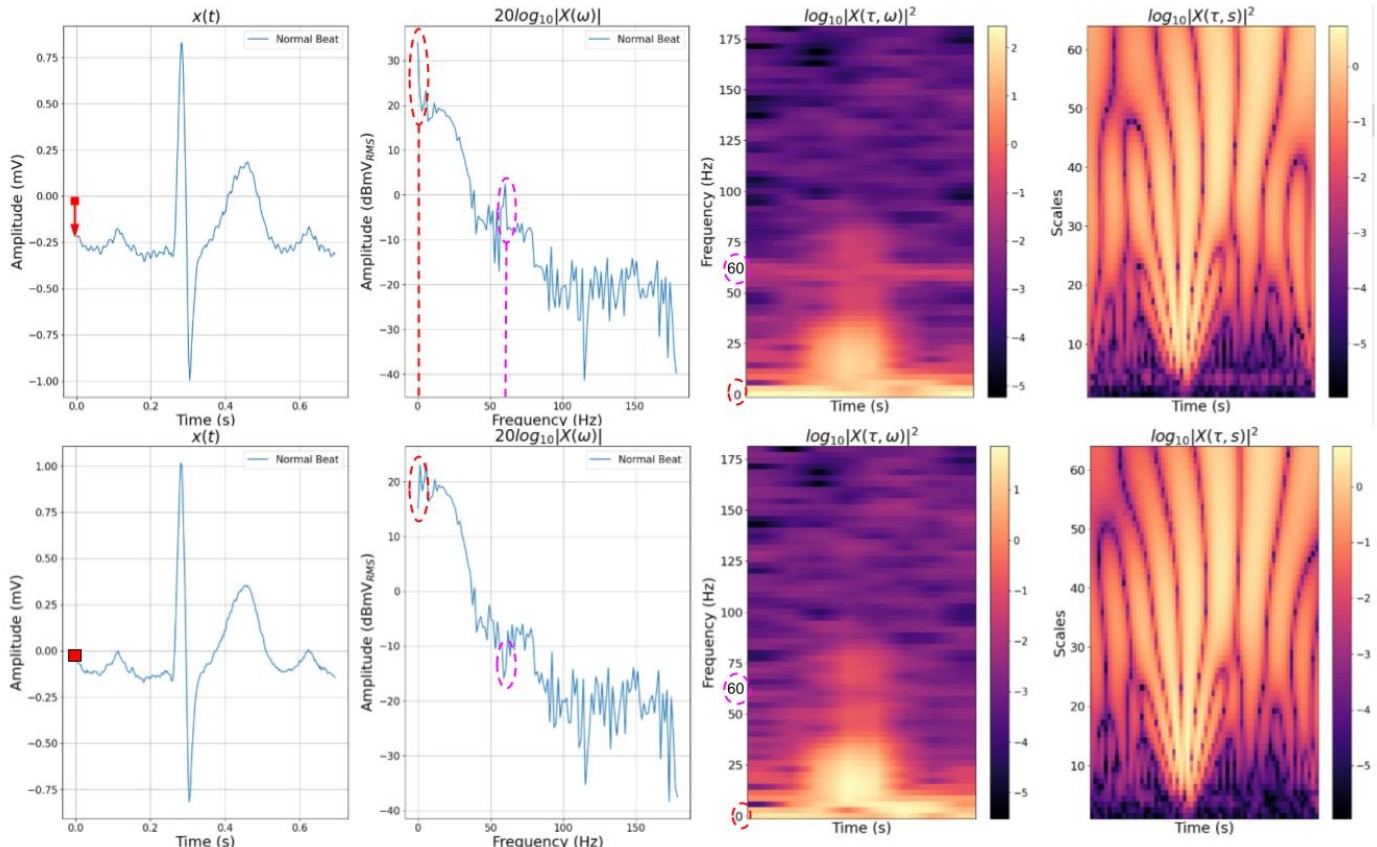

**Figure 1.** Learning representations. From top left to bottom right: time representation of a regular beat corrupted with BW and PLI noises; its frequency representation; its spectrogram representation computed by the STFT using a 128 window size and 75% overlap; its scalogram representation computed by the CWT with a Gauss7 mother wavelet with 64 scales; time representation with BW and PLI noise suppression; frequency representation with BW and PLI noise suppression; the spectrogram representation with BW and PLI noise suppression; and the scalogram representation with BW and PLI noise suppression.

**Table 2.** CNN architecture

| Layer | Kernel Dimension | No. of Filters | Output Shape | No. of Parameters |
|---|---|---|---|---|
| Input | - | - | $1 \times 64 \times 64$ | 0 |
| Conv2D | $3 \times 3$ | 16 | $16 \times 64 \times 64$ | 160 |
| ReLU | - | - | $16 \times 64 \times 64$ | 0 |
| MaxPool2D | $2 \times 2$ | - | $16 \times 32 \times 32$ | 0 |
| Conv2D | $3 \times 3$ | 32 | $32 \times 32 \times 32$ | 4640 |
| ReLU | - | - | $32 \times 32 \times 32$ | 0 |
| MaxPool2D | $2 \times 2$ | - | $32 \times 16 \times 16$ | 0 |
| Conv2D | $3 \times 3$ | 64 | $64 \times 16 \times 16$ | 18,496 |
| ReLU | - | - | $64 \times 16 \times 16$ | 0 |
| MaxPool2D | $2 \times 2$ | - | $64 \times 8 \times 8$ | 0 |
| Dropout | - | - | $64 \times 8 \times 8$ | 0 |
| Dense | - | - | 16 | 65,552 |
| Dense | - | - | 4 | 68 |
| | | | | **Total = 88,916** |

*2.4. Experimental Setup*

2.4.1. Feature Representations

A series of experiments were performed considering both types of 2D features: the power spectrograms, computed by the STFT, and the power scalograms, computed by the CWT.

For the computation of the spectrograms, the parameters experimented with were the window size of the Fourier Transform and the overlap percentage of each window. The values considered for the window size were 32, 64, and 128. For the overlap percentage, the values considered were 75%, 87.5%, and 93.75%. The type of window used was a Hanning window for all of the different setups [23].

For the scalograms, the parameters considered were the mother wavelet and its number of scales, where the choice of the number of scales "$S$" went from 1 to $S$ with a step of 1. The mother wavelets chosen for the experiments were the following: the Morlet wavelet; the second derivative of the Gaussian function, commonly called the Mexican Hat wavelet; and the seventh derivative of the Gaussian function, denoted here as the Gauss7 wavelet for simplicity. The number of scales experimented with was 16, 32, and 64.

We chose the Morlet and Mexican Hat wavelets because of their wide use in the literature considering ECG [24–26]. In regard to the Gauss7 wavelet, we chose this one mainly because we could not find works in the literature that had experimented with it before (the one that came close to it ran experiments up to the sixth derivative of the Gaussian function [27]) and also to check the effects that an asymmetric wavelet would have on the ECG signal.

Figure 2 displays a graph with the map of the wavelet scale values to their respective frequency values considering a sample rate of 360 samples per second. Likewise, the highest and smallest frequency values that each wavelet can represent, with their respective scale values, are displayed in Table 3.

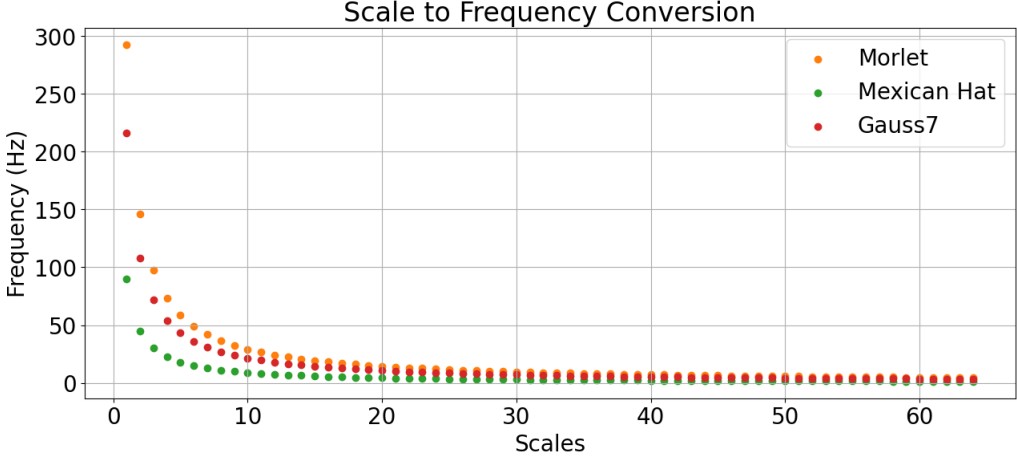

**Figure 2.** Wavelet scale values mapped to their respective frequency values.

**Table 3.** Wavelets smallest and highest frequencies, where "$S$" represents the corresponding scale value.

|  | $S = 1$ | $S = 16$ | $S = 32$ | $S = 64$ |
|---|---|---|---|---|
| **Morlet** | 292.5 Hz | 18.281 Hz | 9.141 Hz | 4.570 Hz |
| **Mexican Hat** | 90.0 Hz | 5.625 Hz | 2.812 Hz | 1.406 Hz |
| **Gauss7** | 216.0 Hz | 13.5 Hz | 6.750 Hz | 3.375 Hz |

In order to compare the effects of filtering the specific types of common ECG noises, four filtering setups were added to the experiments:

- No filter;
- BW filter;
- PLI filter;
- BW + PLI filter

This resulted in 36 configurations for the spectrograms and scalograms, totaling 72 different configurations.

### 2.4.2. Dataset Configurations

Table 4 showcases the dataset split configurations. The database is highly imbalanced, with 90% of the samples belonging to the standard class, which is typical of such a problem.

We opted to use something other than undersampling or oversampling strategies to balance the dataset since this work aimed not to acquire high classification performance on each class. We aimed to study the effects of preprocessing parameter selection considering the same conditions for the dataset and the model, even if the conditions favored classification performance on a specific class and caused performance to deteriorate on another.

**Table 4.** Dataset split configuration.

|  | **N** | **VEB** | **SVEB** | **F** |
|---|---|---|---|---|
| **Train** | 60,946 | 4423 | 1847 | 531 |
| **Validation** | 5224 | 379 | 159 | 45 |
| **Test** | 20,897 | 1516 | 633 | 182 |
| **Total** | 87,067 | 6318 | 2639 | 758 |

After the split, the data in each set went through a Min–Max normalization:

$$NewData := \frac{data - min(set)}{max(set) - min(set)}. \tag{6}$$

In order to validate the model's generalization performance and compute consistent classification metrics, the dataset samples were arranged in 30 different manners, but always maintained the division showcased in Table 4.

This meant that 30 different rounds of experiments were executed on top of the 72 different configurations mentioned in Section 2.4.1, totaling 2160 executions: 1080 executions for the STFT configurations and another 1080 for the CWT configurations.

### 2.4.3. Model Configurations

All of the configurations were fixed regarding the model, including its random initialization. Such an approach aimed to assess only the influence of different dataset arrangements and, of course, the specific learning representations.

It was essential to get conclusive results, as another random configuration on top of the experiments would increase the bias and, therefore, would make it harder to extract conclusions from the preprocessing parameters.

Regarding the model's hyperparameters, the number of epochs chosen was 10. The learning rate was fixed at $1 \times 10^{-3}$, the batch size was 64, an ADAM optimizer was used, and a Cross-Entropy loss function. The models chosen for inference in each round of experiments were the ones that acquired the lowest validation loss during inference on the validation set.

## 3. Results

The criterion adopted to consider one experiment better than the other was the average F1-Score of the predictions in the test set, since this encompasses both the "precision" and "recall" metrics. The accuracy metric may be misleading, especially when the data is highly imbalanced, which was the case in the present study. The computation of the average F1-

Score was accomplished by applying the mean of the F1-Scores across all of the individual classes in the test set, such that

$$\mathrm{Avg.\,F1Score} = \frac{\mathrm{F1Score}(N) + \mathrm{F1Score}(VEB) + \mathrm{F1Score}(SVEB) + \mathrm{F1Score}(F)}{\mathrm{TestSet}}. \quad (7)$$

Figures 3–5 display the F1-Score distribution over the 30 experiments for the Mexican Hat, Gauss7, and Morlet wavelets in the scalogram experiments, respectively. Likewise, Figures 6–8 display the F1-score distribution for the 75%, 87.5% and 93.75% overlap windows for the spectrogram experiments, respectively.

The higher the alphabet letter located on the x-axis in the figures, the lower the average F1-Score. To aid the identification of the experiments and to display the average F1-Scores of the complete set of experiments, Tables 5–10 were generated.

**Table 5.** Mexican hat wavelet experiment identifiers.

| | \multicolumn{12}{c}{**Experiment ID**} |
| | **A** | **B** | **C** | **D** | **E** | **F** | **G** | **H** | **I** | **J** | **K** | **L** |
|---|---|---|---|---|---|---|---|---|---|---|---|---|
| **Scales** | 64 | 32 | 64 | 32 | 16 | 32 | 32 | 64 | 64 | 16 | 16 | 16 |
| **BW Filter** | Yes | Yes | Yes | No | Yes | Yes | No | No | No | No | Yes | No |
| **PLI Filter** | No | Yes | Yes | Yes | Yes | No | No | Yes | No | Yes | No | No |
| **Avg. F1-Score (%)** | 90.1076 | 89.9660 | 89.9559 | 89.7400 | 89.7398 | 89.6619 | 89.6226 | 89.5942 | 89.5159 | 89.4974 | 89.4843 | 89.4533 |

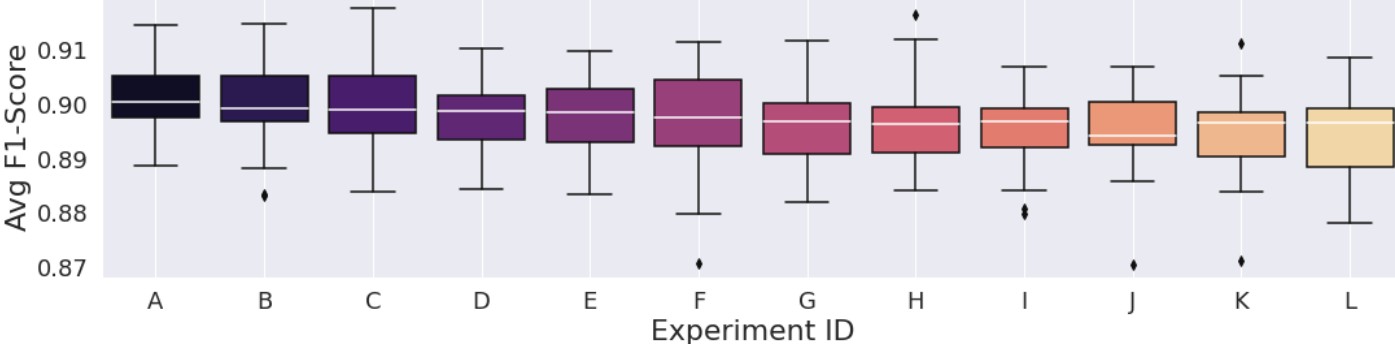

**Figure 3.** Results from the scalograms generated by the Mexican hat Wavelet.

**Table 6.** Gauss7 wavelet experiment identifiers.

| | \multicolumn{12}{c}{**Experiment ID**} |
| | **A** | **B** | **C** | **D** | **E** | **F** | **G** | **H** | **I** | **J** | **K** | **L** |
|---|---|---|---|---|---|---|---|---|---|---|---|---|
| **Scales** | 64 | 64 | 64 | 64 | 32 | 32 | 32 | 32 | 16 | 16 | 16 | 16 |
| **BW Filter** | Yes | Yes | No | No | Yes | Yes | No | No | Yes | Yes | No | No |
| **PLI Filter** | No | Yes | No | Yes | No | Yes | Yes | No | No | Yes | No | Yes |
| **Avg. F1-Score (%)** | 89.6635 | 89.6296 | 89.2273 | 89.0742 | 88.3949 | 88.3348 | 87.8553 | 87.6770 | 83.8375 | 83.5241 | 82.8184 | 82.6222 |

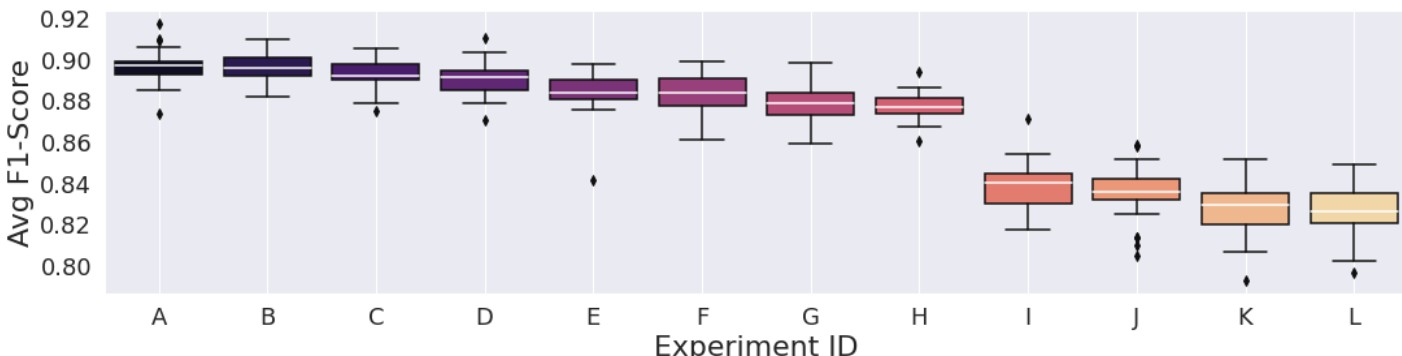

**Figure 4.** Results from the scalograms generated by the gauss7 wavelet.

**Table 7.** Morlet wavelet experiment identifiers

| | Experiment ID | | | | | | | | | | | |
|---|---|---|---|---|---|---|---|---|---|---|---|---|
| | **A** | **B** | **C** | **D** | **E** | **F** | **G** | **H** | **I** | **J** | **K** | **L** |
| **Scales** | 64 | 64 | 64 | 64 | 32 | 32 | 32 | 32 | 16 | 16 | 16 | 16 |
| **BW Filter** | Yes | Yes | No | No | Yes | Yes | No | No | Yes | Yes | No | No |
| **PLI Filter** | No | Yes | No | Yes | No | Yes | Yes | No | No | Yes | No | Yes |
| **Avg. F1-Score (%)** | 88.9853 | 88.8406 | 88.0451 | 87.9009 | 85.5765 | 85.2684 | 84.0790 | 83.9770 | 79.5599 | 79.3363 | 77.8124 | 77.2947 |

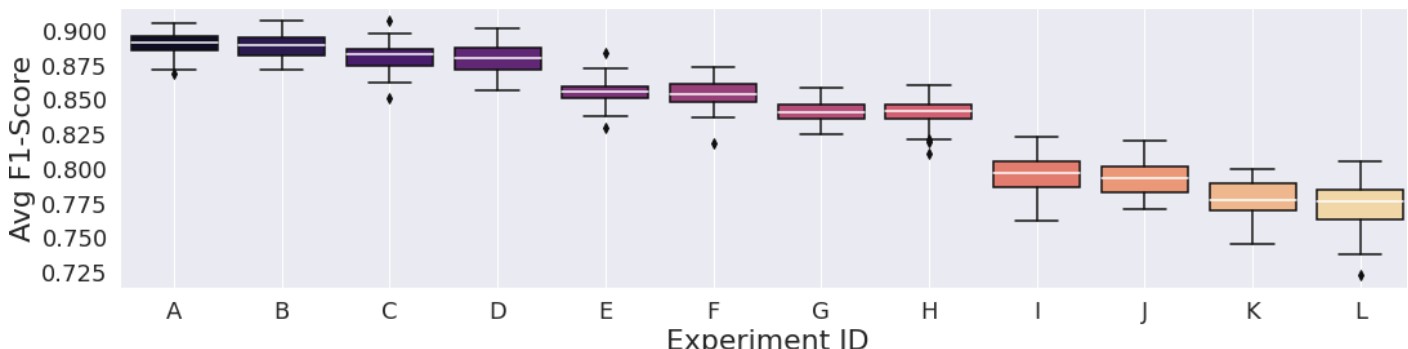

**Figure 5.** Results from the scalograms generated by the morlet Wavelet.

**Table 8.** 75% overlap experiment identifiers

| | Experiment ID | | | | | | | | | | | |
|---|---|---|---|---|---|---|---|---|---|---|---|---|
| | **A** | **B** | **C** | **D** | **E** | **F** | **G** | **H** | **I** | **J** | **K** | **L** |
| **FFT/Window Size** | 64 | 64 | 32 | 32 | 128 | 64 | 128 | 64 | 32 | 32 | 128 | 128 |
| **BW Filter** | Yes | Yes | Yes | Yes | Yes | No | Yes | No | No | No | No | No |
| **PLI Filter** | No | Yes | No | Yes | No | No | Yes | Yes | No | Yes | Yes | No |
| **Avg. F1-Score (%)** | 86.7556 | 86.6069 | 86.2168 | 86.0142 | 84.7389 | 84.3025 | 83.8166 | 83.8125 | 83.7484 | 83.4633 | 81.9681 | 81.7185 |

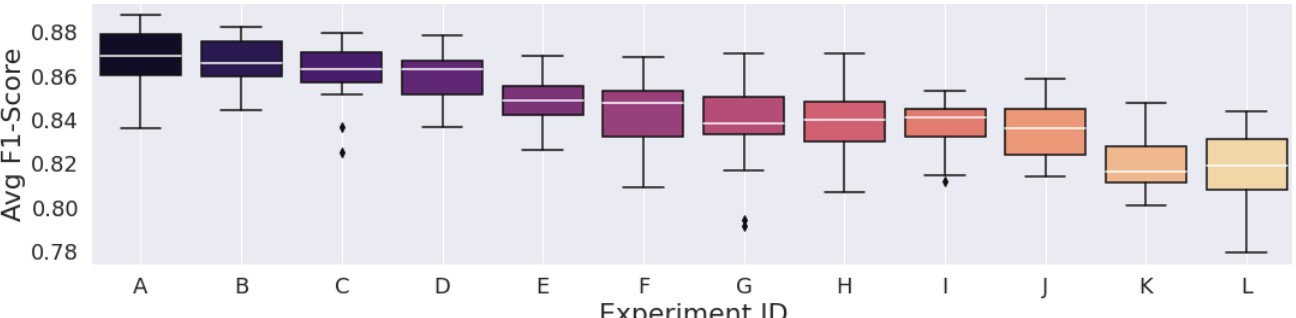

**Figure 6.** Results from the spectrograms generated with 75% window overlap.

**Table 9.** 87.5% overlap experiment identifiers

| | Experiment ID | | | | | | | | | | | |
|---|---|---|---|---|---|---|---|---|---|---|---|---|
| | **A** | **B** | **C** | **D** | **E** | **F** | **G** | **H** | **I** | **J** | **K** | **L** |
| **FFT/Window Size** | 64 | 64 | 128 | 32 | 32 | 128 | 64 | 64 | 128 | 128 | 32 | 32 |
| **BW Filter** | Yes | Yes | Yes | Yes | Yes | Yes | No | No | No | No | No | No |
| **PLI Filter** | No | Yes | No | Yes | No | Yes | No | Yes | No | Yes | No | Yes |
| **Avg. F1-Score (%)** | 87.1210 | 86.9639 | 86.4369 | 86.1286 | 86.0712 | 85.9963 | 84.8144 | 84.7086 | 84.1529 | 84.1522 | 84.0212 | 83.9235 |

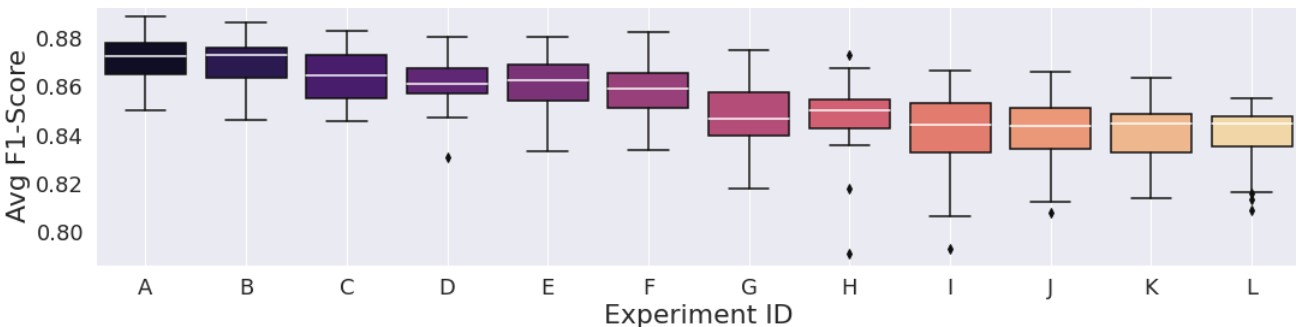

**Figure 7.** Results from the spectrograms generated with 87.5% window Overlap.

**Table 10.** 93.75% overlap experiment identifiers

| | Experiment ID | | | | | | | | | | | |
|---|---|---|---|---|---|---|---|---|---|---|---|---|
| | **A** | **B** | **C** | **D** | **E** | **F** | **G** | **H** | **I** | **J** | **K** | **L** |
| **FFT/Window Size** | 64 | 64 | 32 | 128 | 32 | 128 | 64 | 128 | 128 | 64 | 32 | 32 |
| **BW Filter** | Yes | Yes | Yes | Yes | Yes | Yes | No | No | No | No | No | No |
| **PLI Filter** | Yes | No | No | No | Yes | Yes | Yes | Yes | No | No | No | Yes |
| **Avg. F1-Score (%)** | 86.6833 | 86.6600 | 86.4335 | 86.3896 | 86.1367 | 86.0198 | 84.8327 | 84.7946 | 84.7190 | 84.5660 | 84.1327 | 84.0701 |

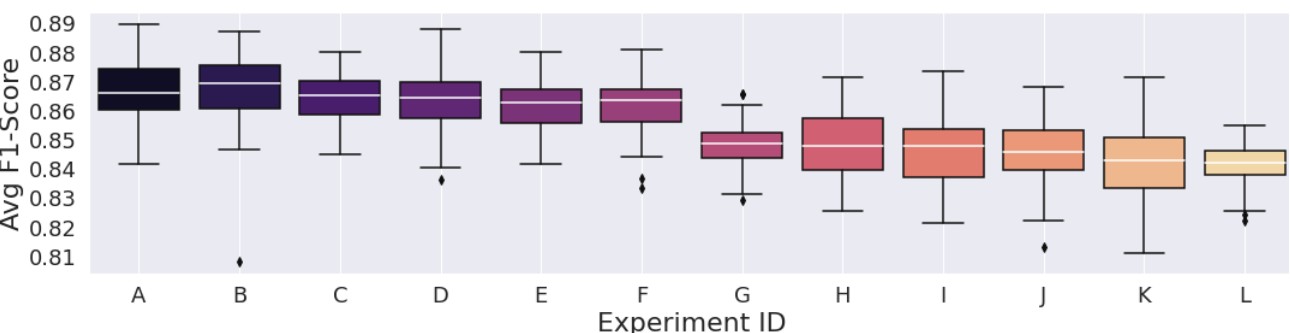

**Figure 8.** Results from the spectrograms generated with 93.75% window Overlap.

## 4. Discussion

The best experiments considered the scalogram as the learning representation input to the model. One of the most notable drawbacks of the STFT approach is that the size of the window $g(t)$ represents a trade-off on time and frequency resolutions [28], in that the larger the window, the better the resolution in the frequency domain but the worse the resolution in the time domain. The smaller the window, the better the resolution in the time domain but the worse the resolution in the frequency domain.

On the other hand, the CWT is considered a more robust technique because, unlike the STFT, which has the trade-off mentioned above, the wavelets in the CWT are convolved on the signal with multiple scales, providing a multi-resolution representation. This is an advantageous property for biological signals, since abrupt frequency variations on the

signals over time and low variations can be mapped simultaneously [29]. Thus, these results were not exactly unexpected.

The best experiment using the STFT, which was experiment "A" in Table 9, achieved an F1-score of 87.12%, which was higher than those of various CWT experiments. Overall, it reached 25th place when considering all 72 experiments.

Regarding the CWT, contrary to the STFT, the model's classification performance was expected to be directly dependent on the number of scales used to represent the wavelets, as the level of information that the scalogram can represent is also directly dependent on this. This can be observed in Tables 6 and 7 that refer to the Gauss7 and Morlet wavelet experiments, respectively. We observed that the lower sets of scales severely deteriorated the classification performance in experiments considering these wavelets.

An interesting finding, however, was that this pattern could not necessarily be observed with the Mexican Hat wavelet, as the results using 32 or even 16 scales were better than some that used 64 scales (Table 5).

Concerning the STFT parameters, we noted that the highest metric, considering the 93.75% overlap, was lower than the highest average F1-Scores of the other overlap experiments. In this peculiar configuration, the spectrograms had high time correlation, as it provided more valued repetition across the STFT time frames, so a high number of similar data points were fed to the convolutions of the Deep Learning model, which should result in the most redundant configuration regardless of window size.

However, the experiment with the lowest average F1-Score was still higher than the other two overlap configurations. Furthermore, the configuration with the lowest time correlation, the 75% overlap, had the lowest average F1-Score among the other two overlap experiments. This led us to believe that, despite such a large number of redundant values, the network seemed to have a more steady learning procedure in the configuration with time-dependence.

Having said all that, the STFT's parameter search and selection proved not to be intuitive. The largest FFT combined with the most significant overlap percentage did not generate the best results. Of course, this was expected because of STFT's constant resolution, which also results in time and frequency resolution trade-offs. The choice of parameters that result in the best classification metrics highly depends on the target signal type.

Concerning the CWT parameters, the choice of mother wavelet played an important role in the resulting learning representations fed to the network. When we analyzed the wavelets overlapped with a generic Normal ECG beat from the dataset in Figure 9, we noticed that the wavelets overlapped with important ECG points, particularly the QRS complex features. The time shift from these wavelets was chosen to coincide peaks with this particular ECG R-peak to facilitate comprehension.

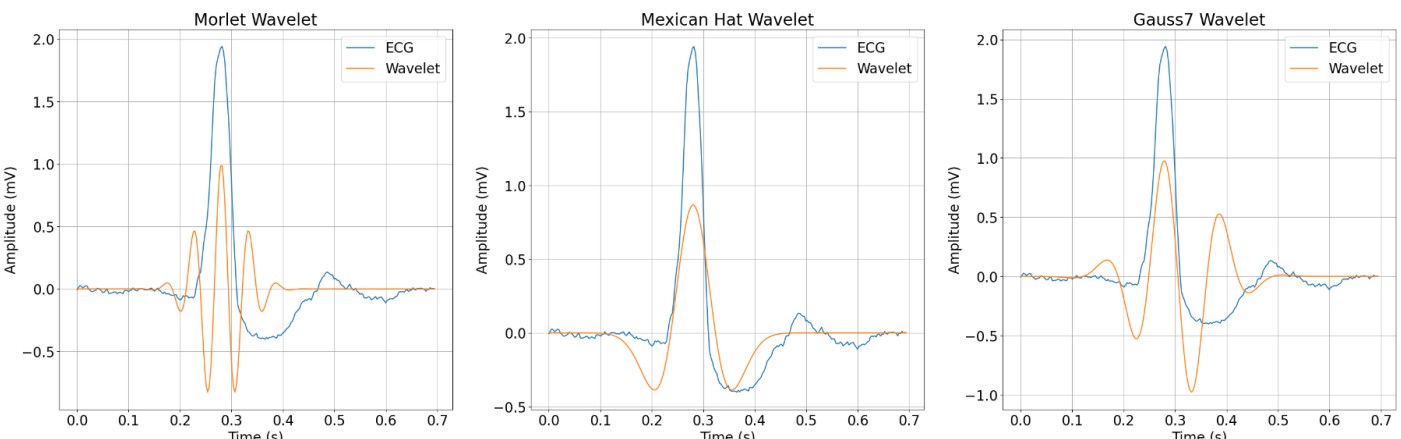

**Figure 9.** Overlap visualization of an ECG beat and the chosen wavelets.

When looking at the figure, one may note that, despite the Morlet wavelet having some overlapping points with the ECG features, it was also the one wavelet that least represented the ECG shape. This may explain why the wavelet achieved the worst performance overall when considering the evaluation metrics. As for the Gauss7 wavelet, we chose to experiment with it because its shape is that of an ECG shape, especially considering its asymmetric aspect. Still, it did not reach the overall performance of the Mexican Hat wavelet, which had fewer vanishing moments and performed better with most of the filtering and scale configurations.

If we consider that the most relevant information of an ECG is found at its QRS complex, the results may be explained, because the Mexican Hat wavelet had a single peak and it overlapped with the ECG R-peak at a generic time location $t_0$. At that same time, no other peaks in the wavelet overlapped with any part of the signal, and these "extra" peaks could possibly, in turn, have createc a misleading scalogram representation of the ECG. Considering the time dimension, this may be one reason why the Gauss7 and Morlet wavelets performed worse than the Mexican Hat wavelet.

It is, however, easier to interpret why the Mexican Hat wavelet was better than the other two by checking the frequency spectrum of an arbitrary Normal ECG beat (the same used in Figure 9) and comparing it to the natural frequency spectrum of the mother wavelets, as in Figure 10. The ECG waveforms had more energy in the lowest frequencies and, by referring to the conversions of these wavelet scales in Figure 2 and also to Table 3, one can see that the Mexican Hat wavelet was more capable of representing smaller frequencies than the other two wavelets.

This also may explain why the Morlet wavelet with 16 scales severely deteriorated the classification performance compared to its other scale values, as the lowest frequency it could represent was above 18 Hz, while the Gauss7 wavelet could represent slightly smaller values and, as for the Mexican Hat, it could represent considerably smaller values.

In the frequency spectrum, one can see that the Mexican Hat mother wavelet's frequency values carrying more energy overlapped with those from the ECG beat in the lowest frequencies. It is important to remember that the CWT performs a convolutional operation, and a convolution in the frequency spectrum is multiplication, so the fact that the values with more energy in the frequency domain overlap means there are higher values resulting after the multiplication operation. Such an effect is displayed in Figure 11.

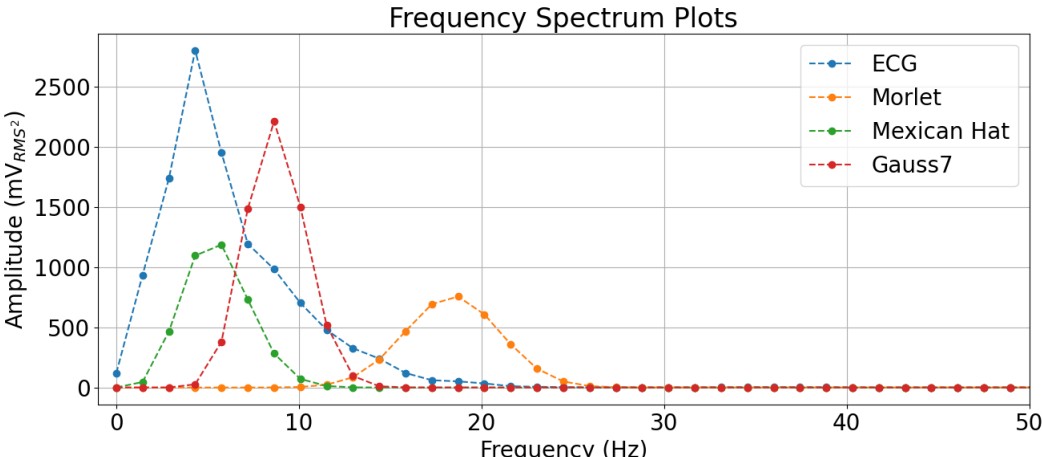

**Figure 10.** Visualization of the Frequency Spectrum of an ECG Beat and the Mother Wavelets.

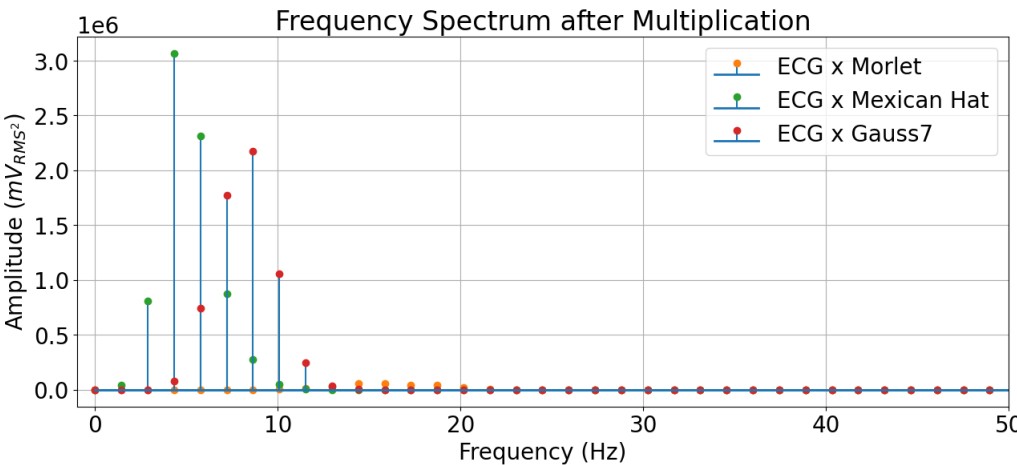

**Figure 11.** Visualization of the frequency spectrum of an ECG beat multiplied by the mother wavelets.

The Mexican Hat wavelet, therefore, highlights the frequencies in the ECG that carry more energy. The Gauss7 wavelet, in turn, highlights the subsequent frequencies, even though these are not the ones with the highest energies (despite carrying a significant amount), which explains why this wavelet reached second place in the results. As for the Morlet wavelet, it is evidenced again through the frequency plots that this wavelet did not carry enough energy in the frequencies that composed the ECG, which explained its overall poor performance in the classification experiments, compared to the others.

We can also observe that the F1-Score distribution of the experiments that considered the scalograms generated by the Mexican Hat wavelet was a lot more "stable" than those of the other wavelets. As pointed out, reducing scales in the Gauss7 and Morlet wavelets considerably deteriorated classification performance compared to their highest results. In contrast, all the F1-Scores achieved by the Mexican Hat wavelet, including the experiment with the lowest set of scales, were still higher than 89%.

Overall, we confirmed, with these results, that the CWT is more efficient regarding classification performance than the STFT in classifying arrhythmias. It is also simpler and more intuitive to experiment with, as it has fewer parameters to choose from than the STFT, and their respective effects on the classification performance are predictable.

However, since we used the multi-resolution property and performed many convolutions for the scale representations, the computational complexity was naturally more considerable than that of the STFT. The STFT performs only one convolution plus the Fourier Transform in each frame, so it is essential to consider this when designing a low-cost system.

The use of a BW filter proved to be very necessary in order to obtain higher classification performances in most experiments. This was expected, as BW is the noise with the most significant influence on a signal's energy, as discussed in Section 2.2.4. However, the same did not apply to the PLI noise. Most experiments that used only the BW filter had a higher classification performance than those that used both filters. Even some that used no filters achieved better performance than those using only the PLI filter.

As explained in Section 2.2.3, PLI is a noise that usually affects the 50 or 60 Hz frequencies and, as discussed in the previous analysis. As evidenced in the frequency plots in Figures 1–10, the ECG signals barely carried any relevant information beyond the 40 or 50 Hz frequencies. Thus, the 50/60 Hz notch filter did not significantly affect the feature extraction accuracy for arrhythmia classification. The Notch filter in this application was also used in other works, such as [30,31].

Concerning the BW noise, it affected the lowest frequency values in the ECG, so a high-pass filter was required to cancel such noise. The Butterworth filter was chosen for this task, and has been used in other works with different orders to cancel high- or low-frequency noises. The filter order chosen affects the frequency response, and we chose

a second-order filter to yield less filter complexity in contrast to works such as [31] that used a third-order Butterworth filter for low frequencies.

## 5. Conclusions and Future Works

With data generated from a consistent and reproducible experimental methodology, this paper elucidates the importance of the selection of signal processing feature extraction techniques, the parameter selection that comes with them, and the application of noise filters when using computational models for the classification of arrhythmias.

We show that the CWT works better as a spectral transform for classifying arrhythmia than the STFT, primarily due to its multi-resolution property and the flexibility provided by wavelet functions with different frequencies. Even though its computational cost is higher than the STFT, the Mexican Hat wavelet did not require the highest set of scales to reach a place in the top 5 best experiments.

The fifth best experiment, when considering the F1-Score as the classification metric, came from a scalogram that was generated from a Mexican Hat wavelet with the lowest set of scales chosen in the experiments, which was 16. This is explained by this wavelet's capacity of highlighting information at low frequencies, which are the particular ones that carry most ECG information. This information is helpful, as we observe that it may be possible to design a system with even lower computational complexity by not using most of the wavelet scales and maybe achieving better classification metrics using data balancing strategies, which were not used in the present work.

We used a simple model to achieve such classification metrics, as the CNN we used had only 88,916 parameters.

We intend, in future work, to perform a deeper analysis of the learning representations (both scalograms and spectrograms) by using methods such as the Kullback–Leibler Divergence (i.e., Relative Entropy) [32] to check precisely where there is valuable information in the representations that may aid the CNN to differentiate between samples. Thus, by identifying regions with valuable information, we could use fewer features to perform the classification, achieving similar performances with a lower computational burden.

This opens many doors of possibilities, as we may direct the convolutions to specific points and cut computational complexity even further during training and inference. It might also lead us to using a CNN with a lower set of parameters if the classification performance can be increased by a significant margin.

In the future, it will be essential to include more metrics related to computational cost, other than the number of model parameters, such as the number of FLOPs and model size in bytes, to embed the whole system in a device.

**Author Contributions:** Conceptualization, R.H., R.M. and C.B.-F.; methodology, R.M. and C.B.-F.; software, R.H.; validation R.H., R.M. and C.B.-F.; writing—original draft preparation, R.H.; writing—review and editing, R.M. and C.B.-F.; supervision, C.B.-F. All authors have read and agreed to the published version of the manuscript.

**Funding:** This research received no external funding.

**Institutional Review Board Statement:** Not applicable.

**Informed Consent Statement:** Not applicable.

**Data Availability Statement:** The dataset used in the experiments is a publicly available dataset called MIT-BIH.

**Acknowledgments:** We would like to thank the Post-Graduate Program in Computer Engineering (PPGEC) of the University of Pernambuco.

**Conflicts of Interest:** The authors declare no conflict of interest.

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
