# Peer review of "Preprocessing Selection for Deep Learning Classification of Arrhythmia Using ECG Time-Frequency Representations"

_technologies, doi:10.3390/technologies11030068_

Round 1

Reviewer 1 Report

1.  First citation [1] refers to Cardio-vascular disease (CVD) related to alcohool consumption, which does not make it the leading cause of mortality worldwide, except if most of the patients drink a significant amount of alcohool. This reference could be improved.

2. It is citing (line 76) that the author want to fill the gap of "the lack of an explainable parameter selection tha tresult in the learning representation of ECG signals." In §Discussion or Conclusion, an explanation of how the current manuscript is reaching that goal could be more explicitely disclosed.

3. Since one conclusive hypothesis is related to the filtering of PLI , it could be necessary to citemore in detail some features of this filter.

4. The main concerns for us, is the lack of clear definition of average F1-score. What is the range of this average F1-score results? The lack of results using the validation database lead to the question : Does average F1-score encompasses learning + test + validation database results or different types of ECG rhythms ? , it is not described whether the results are coming from which database (learn, test, validation).

In the same manner, it is not clear (or not clear enough) whether the learning phase of each test has been perfomed with a combinatory application of the filters (BLW, PLI) or if filters are applied afterwards.

Author Response

Reviewer 1:

  1. First citation [1] refers to Cardio-vascular disease (CVD) related to alcohool consumption, which does not make it the leading cause of mortality worldwide, except if most of the patients drink a significant amount of alcohool. This reference could be improved.

R: Certainly. We have added the following comment: “especially considering people who deal with problems concerning alcohol consumption.”

  1. It is citing (line 76) that the authors want to fill the gap of "the lack of an explainable parameter selection that results in the learning representation of ECG signals." In §Discussion or Conclusion, an explanation of how the current manuscript is reaching that goal could be more explicitly disclosed.

R: Thanks for the remark, and sorry for not giving this topic enough focus in the first version. On page 13, we added a series of paragraphs starting at “As far as explainability goes….”  that discuss the effects of the transform parameter choice for both the STFT concerning the overlap windows and their possible effects of time-dependency in the classification performance; and the CWT concerning the wavelets and how their natural frequencies make them appropriate or not for the use in ECG signals.

Here is the text:

“As far as explainability goes, concerning the STFT parameters, we note that the highest metric considering the 93.75% overlap was lower than the highest average F1-Scores of the other overlap experiments. In this configuration, in particular, the spectrograms have high time correlation, as it provides more value repetition across the STFT time frames, so a high number of similar data points are fed to the convolutions of the Deep Learning model, which should result in the most redundant configuration regardless of the window size.

However, its experiment with the lowest average F1-Score is still higher than the other two overlap configurations. Moreover, the configuration with the lowest time correlation, the 75% overlap, had the lowest average F1-Score among the other two overlap experiments. It leads us to believe that, despite such a large number of redundant values, the network has a more steady learning procedure from the most time-dependent configuration, which suggests a higher time dependency in the ECG signals than a frequency dependency.

That said, the STFT's parameter search and selection proved not intuitive, as the largest FFT combined with the most significant overlap percentage does not generate the best results. Of course, this is expected because of STFT's constant resolution, which also results in time and frequency resolution trade-offs. The choice of parameters that result in the best classification metrics will depend highly on the target signal type.

Concerning the CWT parameters, the choice of mother wavelet played an essential role in the resulting learning representations fed to the network. When we analyze the wavelets overlapped with a generic Normal ECG beat from the dataset in Figure 9, it is possible to notice that the wavelets overlap with important ECG points, particularly the QRS complex features. The time shift from these wavelets was chosen to coincide their peaks with this particular ECG R-peak to facilitate comprehension.

When looking at the figure, one may note that, despite the Morlet wavelet having some overlapping points with the ECG features, it is also likely that the one wavelet representing the ECG shape the least. It may explain why the wavelet achieved the worst performance overall when considering the evaluation metrics. As for the Gauss7 wavelet, we chose to experiment with it because its shape reminds that of an ECG shape, especially considering its asymmetric aspect. Still, it has yet to reach the overall performance of the Mexican Hat wavelet, which has fewer vanishing moments and overperforms with most of the filtering and scale configurations.

Let us consider that the most relevant information about an ECG is found at its QRS complex. Such a thing may be explained, as the Mexican Hat wavelet has a single peak and overlaps with the ECG R-peak at a generic time location ${t_0}$. At that exact time location, no other peaks in the wavelet overlap with any part of the signal, and these "extra" peaks could, in turn, create a misleading scalogram representation of the ECG. Considering the time dimension, this is one of the reasons why the Gauss7 and Morlet wavelets achieve worse performance than the Mexican Hat wavelet.

However, it is easier to interpret why the Mexican Hat wavelet was better than the other two by checking the frequency spectrum of an arbitrary Normal ECG beat (the same used in Figure 9) compared to the natural frequency spectrum of the mother wavelets in Figure 10. It is evident that the ECG waveforms have more energy in the lowest frequencies, and by referring to the conversions of these wavelet scales in Figure 2 and also to Table 3, one can see that the Mexican Hat wavelet is more capable of representing more minor frequencies than the other two wavelets.

It also may explain why the Morlet wavelet with 16 scales had a severe deterioration in classification performance compared to its other scale values, as the lowest frequency it can represent is above 18 Hz, while the Gauss7 wavelet can represent slightly smaller values and, as for the Mexican Hat, considerably smaller values.

Also, in the frequency spectrum, one can see that the Mexican Hat mother wavelet's frequency values that carry more energy overlap with those from the ECG beat in the lowest frequencies. It is important to remember that the CWT performs a convolution operation, and a convolution in the frequency spectrum is multiplication, so the fact that the values with more energy in the frequency domain overlap means there will result in higher values after the multiplication operation. Such an effect is displayed in Figure 11.

The Mexican Hat wavelet, therefore, highlights the frequencies in the ECG that carry more energy. The Gauss7 wavelet, in turn, highlights the subsequent frequencies, even though those are not the ones with the highest energies (despite carrying a significant amount), which explains why this wavelet reached second place in the results. As for the Morlet wavelet, it is evidenced again through the frequency plots that this wavelet does not carry enough energy in the frequencies that compose the ECG, which explains its overall poor performance in the classification experiments compared to the others."

  1. Since one conclusive hypothesis is related to the filtering of PLI , it could be necessary to cite more in detail some features of this filter.

R: We have added more information concerning both filters with the following text on page 15:

“As explained in 2.2.3, PLI is a noise that usually affects the 50 or 60 Hz frequencies and, as discussed in the previous analysis, evidenced in the frequency plots in Figures \ref{learning rep} and \ref{spectrums}, the ECG signals barely carry any relevant information beyond the 40 or 50 Hz frequencies, so it is natural to conclude that such filter is not necessary for proper arrhythmia classification. The Notch filter in this application has also been used in other works such as \cite{bai2004adjustable} and \cite{jagtap2012impact}. 

The BW noise affects the lowest frequency values in the ECG, so a high-pass filter is required to cancel such noise. The Butterworth filter was chosen for this task, and it has been used in other works with different orders to cancel high- or low-frequency noises. The filter order chosen affects the frequency response, and we chose a second-order filter to yield less filter complexity in contrast to works such as \cite{jagtap2012impact} that have used a third-order Butterworth filter for the low frequencies.”

  1. The main concerns for us, is the lack of clear definition of average F1-score. What is the range of this average F1-score results? The lack of results using the validation database lead to the question : Does average F1-score encompasses learning + test + validation database results or different types of ECG rhythms ? , it is not described whether the results are coming from which database (learn, test, validation).

In the same manner, it is not clear (or not clear enough) whether the learning phase of each test has been perfomed with a combinatory application of the filters (BLW, PLI) or if filters are applied afterwards.

R: We realized we had not explained much about the evaluation metric, and we see that what we mean by “Avg " can be confusing. F1-Score” Sorry about that. What we mean by that is that we are computing the mean of the F1-Scores across all classes in the test set. The validation set is used solely for computing the validation loss, which in this case, we use the Categorical Cross Entropy, and this measure is used only for choosing the best model checkpoint for inference in the test set. We added some extra information about this methodology in the text on page 9, where we include an equation for the Avg F1-Score metric and added the following text:

“The computation of the average F1-Score was done by applying the mean of the F1-Scores across all of the individual classes in the test set, such that…” and the equation follows the statement.

 We also crossed out a word in the last paragraph of 2. Materials and Methods where we explicit the “inference on the validation set.” 

Concerning filters, the training phase was performed after each filtering configuration cited in the work was applied to the signals. We have added the following text on page 3, right after we mention the preprocessing steps: “The signals go through each of these steps, in the order that they are listed, before being fed to the model, which is explained in subsection 2.3.”

Hopefully, it is more clear now.

Reviewer 2 Report

The idea motivating the paper is very sound and timely. It is the reviewer's opinion that preprocessing and feature extraction are rather neglected aspects in machine learning, therefore this work was approached with genuine interest.

For this very reason, however, I found some disappointing aspects in the work, that I hope the authors will be able to improve upon in a revised version of their manuscript.

My feeling is that the work suffers from a somewhat "black box" approach to the choice of preprocessing approaches, even though these are the object of the paper. By necessity, preprocessing is very much application-oriented, therefore some consideration must be given to the main features of the analyzed waveform, in this case the ECG waveform corresponding to a single heart beat.

At 250 samples and 360 Hz sampling rate, this is slightly less than 700 ms long, although some features (particularly the R peak, that is relevant to arrythmia) are much shorter. In the frequency domain, plots provided by the authors show that frequency content tends to be negligible beyond 30-40 Hz.

3. Results - Comments regarding the STFT:

1) a full-length (250-sample) Fourier transform would yield a frequency step of 1.44 Hz, whereas selected window lengths provide increasingly larger steps of 2.81 Hz, 5.625 Hz and 11.25 Hz. Thus, most useful spectral information in concentrated in very few Fourier coefficients, suggesting that a large part of the spectrogram is almost useless. This should be checked by the authors, since in this case the STFT feature set could be reduced to significantly less than 64x64

2) the choice of overlap values is not motivated. In particular, 93.75% overlap, that can be more esily understood as a 2-sample shift  for each of the 32-sample STFT segments, suggests that spectrogram values might be strongly correlated over time. This should be discussed in connection with the convolutional operations within DL, as it may suggest once again that some information provided to the algorithm is redundant. In 4. Discussion you remark that "The STFT parameter search and selection proved not intuitive", which might perhaps be related.

3) a 64-sample STFT is seemingly the best choice, corresponding to a frequency step of 5.625 Hz. Although lengths that are not expressed in powers of 2 as sample number are not usually considered for efficient computation, the authors might wish to investigate whether a slightly different length  (e.g., 60 or 70 samples) is still close to the best performance. This means ensuring that the 64-sample length is really an optimum, rather than a "lucky hit" in a randomly fluctuating set of results.

3. Results - Comments regarding the wavelet transform

1) please clarify what you mean by "number of scales". I take it that number of scales = 64 means that scale numbers are 1, 2, 4, 8, 16, 32, 64

2) following equation (4) it is clear that the scalogram is considering lower frequencies at higher scale numbers, but you do not provide actual values. I suspect that there is a significant mismatch between the set of features represented by a scalogram, compared to a spectrogram. If this is so, possibly as a consequence of your basic choice of considering single ECG segments of 250 samples, this aspect should be discussed in the interest of a fair comparison.

3) the choice of mother wavelet is an additional option in favour of the wavelet transform. One feels that results are significantly influenced by the shape of the mother wavelet and how it relates to the basic ECG waveform shape. Since your work is concerned with preprocessing, this should be given consideration. As you note in 4. Discussion, with the Mexican Hat wavelet the dependence on the maximum scale value is almost irrelevant, whereas it is rather significant for the other two wavelets, which may point to something in this regard.

Minor remarks:

2.2.1 Beat segmentation - you do not specify this, so I assume you are using R-peak positions taken directly from MIT-BIH annotations. If so, it should be stated.

2.2.4, page 5 - dimension of 64x64 - please clarify: does this mean a common 64x64 grid was employed for all spectrograms and scalograms? That is, 4096 data values per spectrogram/scalogram?

2.2.4, page 5 - annulment ... possibly "cancellation"?

2.2.4, page 6 - spectrum figures are strictly not amplitudes and log (mV^2) is a rather odd choice, as far as units are concerned. Since your values are nevertheless derived from voltage measurements (the ECG traces) I would suggest the use a better suitable logaritmic unit for *amplitude*, namely, dBV = 20 x log_10 (voltage_RMS) - the conversion is rather straightforward if you start from the values of Fourier coefficients

4. Discussion, page 12 - I find it rather hard to believe that a notch filter at 60 Hz does suppress *any* significant information "useful to differentiate cardiac rythms". Components that are over two orders of magnitude less than the most significant ones may at most affect subtle aspects of the waveform shape, but I'd expect them to be irrelevant to arrythmia.

5. Conclusion and Future Works - "CWT is a more robust signal processing technique" - what do you mean by "more robust"? This sounds too generic and might be inappropriate.

At the end of a rather long list of remarks I do encourage you to carry on, as the basic idea is interesting.

Author Response

Reviewer 2:

The idea motivating the paper is very sound and timely. It is the reviewer's opinion that preprocessing and feature extraction are rather neglected aspects in machine learning, therefore this work was approached with genuine interest.

For this very reason, however, I found some disappointing aspects in the work, that I hope the authors will be able to improve upon in a revised version of their manuscript.

My feeling is that the work suffers from a somewhat "black box" approach to the choice of preprocessing approaches, even though these are the object of the paper. By necessity, preprocessing is very much application-oriented, therefore some consideration must be given to the main features of the analyzed waveform, in this case the ECG waveform corresponding to a single heart beat.

At 250 samples and 360 Hz sampling rate, this is slightly less than 700 ms long, although some features (particularly the R peak, that is relevant to arrythmia) are much shorter. In the frequency domain, plots provided by the authors show that frequency content tends to be negligible beyond 30-40 Hz.

  1. Results - Comments regarding the STFT:

1) a full-length (250-sample) Fourier transform would yield a frequency step of 1.44 Hz, whereas selected window lengths provide increasingly larger steps of 2.81 Hz, 5.625 Hz and 11.25 Hz. Thus, most useful spectral information in concentrated in very few Fourier coefficients, suggesting that a large part of the spectrogram is almost useless. This should be checked by the authors, since in this case the STFT feature set could be reduced to significantly less than 64x64

R: The reviewer is correct, and we plan to discuss this topic further in future works. We plan to quantitatively analyze the most helpful information in each representation (either spectrogram or scalogram). It should show numerically that the most helpful information for categorizing the ECG spectrograms is shown only in very few areas of the 64x64 representation. We had already highlighted such an idea in the 5. In the conclusion and Future Works section, we could use tools such as the Kullback Leibler Divergence to perform such analysis. The text is the following:

“ We intend in future works to perform a deeper analysis of the learning representations (both scalograms and spectrograms) by using methods such as the Kullback-Leibler Divergence (\textit{i.e.} Relative Entropy) \cite{van2014renyi} to check precisely where there is valuable information in the representations that may aid the CNN to differentiate between samples. Thus, by identifying those regions with valuable information, we could use fewer features to perform the classification, achieving similar performances with a lower computational burden. ”

2) the choice of overlap values is not motivated. In particular, 93.75% overlap, that can be more esily understood as a 2-sample shift  for each of the 32-sample STFT segments, suggests that spectrogram values might be strongly correlated over time. This should be discussed in connection with the convolutional operations within DL, as it may suggest once again that some information provided to the algorithm is redundant. In 4. Discussion you remark that "The STFT parameter search and selection proved not intuitive", which might perhaps be related.

R: Good remark. We have attempted to discuss this in the text on page 13. We added the following text: 

"As far as explainability goes, concerning the STFT parameters, we note that the highest metric considering the 93.75% overlap was lower than the highest average F1-Scores of the other overlap experiments. In this configuration, in particular, the spectrograms have high time correlation, as it provides more value repetition across the STFT time frames, so a high number of similar data points are fed to the convolutions of the Deep Learning model, which should result in the most redundant configuration regardless of the window size.

However, its experiment with the lowest average F1-Score is still higher than the other two overlap configurations. Furthermore, the configuration with the lowest time correlation, the 75% overlap, had the lowest average F1-Score among the other two overlap experiments. It leads us to believe that, despite such a large number of redundant values, the network seems to have a more steady learning procedure from the most time-dependent configuration, which may suggest a higher time dependency in the ECG signals than a frequency dependency.

That said, the STFT's parameter search and selection proved not intuitive, as the largest FFT combined with the most significant overlap percentage does not generate the best results. Of course, this is expected because of STFT's constant resolution, which also results in time and frequency resolution trade-offs. The choice of parameters that result in the best classification metrics will depend highly on the target signal type."

3) a 64-sample STFT is seemingly the best choice, corresponding to a frequency step of 5.625 Hz. Although lengths that are not expressed in powers of 2 as sample number are not usually considered for efficient computation, the authors might wish to investigate whether a slightly different length  (e.g., 60 or 70 samples) is still close to the best performance. This means ensuring that the 64-sample length is really an optimum, rather than a "lucky hit" in a randomly fluctuating set of results.

R: Thank you for the suggestion. We shall perform such a more refined analysis in a future work.

  1. Results - Comments regarding the wavelet transform

1) please clarify what you mean by "number of scales". I take it that number of scales = 64 means that scale numbers are 1, 2, 4, 8, 16, 32, 64

R: Not really. When we say we have chosen, for example, 32 scales, we mean the scale numbers are “1, 2, 3, 4, …, 32”. The scales are varied with a step of 1. It was clarified in the text on page 7 as “... the choice of the number of scales ‘S’ goes from 1 to S with a step of 1”.

2) following equation (4) it is clear that the scalogram is considering lower frequencies at higher scale numbers, but you do not provide actual values. I suspect that there is a significant mismatch between the set of features represented by a scalogram, compared to a spectrogram. If this is so, possibly as a consequence of your basic choice of considering single ECG segments of 250 samples, this aspect should be discussed in the interest of a fair comparison.

R: Thanks for the comment. We uploaded a plot of scale to frequency conversion (Figure 2) and a table that showcases the lowest frequency values that the chosen wavelets can represent with their respective chosen scale values (Table 3), and we added the following text that refers to them on page 7:

“Figure 2 displays a graph with the map of the wavelet scale values to their respective frequency values considering the sample rate of 360 samples per second. The highest frequency values (the ones that correspond to the scale values of 1) of the Morlet, Mexican Hat, and Gauss7 wavelets are 292.5 Hz, 90 Hz, and 216 Hz. Likewise, the smallest frequency values that each wavelet can represent with their respective largest scale values are displayed in Table 3.”

 We also discussed further in the 4. Discussion section on how these wavelets may be helping classify arrhythmia at a frequency level and the text for this discussion is mentioned in the following remark.

3) the choice of mother wavelet is an additional option in favour of the wavelet transform. One feels that results are significantly influenced by the shape of the mother wavelet and how it relates to the basic ECG waveform shape. Since your work is concerned with preprocessing, this should be given consideration. As you note in 4. Discussion, with the Mexican Hat wavelet the dependence on the maximum scale value is almost irrelevant, whereas it is rather significant for the other two wavelets, which may point to something in this regard.

R: Very relevant comment, thanks. We discussed more in-depth in the text added to the 4.Discussion section on how the wavelets waveform shapes may aid in dealing with the ECG waveforms and the frequency spectrum level, facilitating the understanding process. For that matter, we also included Figure 9 (analysis of wavelet and ECG waveforms), Figure 10 (frequency spectrum plots from the wavelets and ECG), and Figure 11 (plots of the frequency spectrum multiplication). The text added is as follows:

"Concerning the CWT parameters, the choice of mother wavelet played an essential role in the resulting learning representations fed to the network. When we analyze the wavelets overlapped with a generic Normal ECG beat from the dataset in Figure 9, it is possible to notice that the wavelets overlap with important ECG points, particularly the QRS complex features. The time shift from these wavelets was chosen to coincide their peaks with this particular ECG R-peak to facilitate comprehension.

When looking at the figure, one may note that, despite the Morlet wavelet having some overlapping points with the ECG features, it is also likely that the one wavelet representing the ECG shape the least. It may explain why the wavelet achieved the worst performance overall when considering the evaluation metrics. As for the Gauss7 wavelet, we chose to experiment with it because its shape reminds that of an ECG shape, especially considering its asymmetric aspect. Still, it has not reached the overall performance of the Mexican Hat wavelet, which has fewer vanishing moments and overperforms with most of the filtering and scale configurations.

Let us consider that the most relevant information about an ECG is found at its QRS complex. Such a thing may be explained, as the Mexican Hat wavelet has a single peak and overlaps with the ECG R-peak at a generic time location ${t_0}$. At that exact time location, no other peaks in the wavelet overlap with any part of the signal, and these "extra" peaks could, in turn, create a misleading scalogram representation of the ECG. Considering the time dimension, this may be one reason why the Gauss7 and Morlet wavelets perform worse than the Mexican Hat wavelet.

However, it is easier to interpret why the Mexican Hat wavelet was better than the other two by checking the frequency spectrum of an arbitrary Normal ECG beat (the same used in Figure 9) compared to the natural frequency spectrum of the mother wavelets in Figure 10. It is evident that the ECG waveforms have more energy in the lowest frequencies, and by referring to the conversions of these wavelet scales in Figure 2 and also to Table 3, one can see that the Mexican Hat wavelet is more capable of representing more minor frequencies than the other two wavelets.

It also may explain why the Morlet wavelet with 16 scales had a severe deterioration in classification performance compared to its other scale values, as the lowest frequency it can represent is above 18 Hz, while the Gauss7 wavelet can represent slightly smaller values and, as for the Mexican Hat, considerably smaller values.

Also, in the frequency spectrum, one can see that the Mexican Hat mother wavelet's frequency values that carry more energy overlap with those from the ECG beat in the lowest frequencies. It is important to remember that the CWT performs a convolution operation, and a convolution in the frequency spectrum is multiplication, so the fact that the values with more energy in the frequency domain overlap means there will result in higher values after the multiplication operation. Such an effect is displayed in Figure 11.

The Mexican Hat wavelet, therefore, highlights the frequencies in the ECG that carry more energy. The Gauss7 wavelet, in turn, highlights the subsequent frequencies, even though those are not the ones with the highest energies (despite carrying a significant amount), which explains why this wavelet reached second place in the results. As for the Morlet wavelet, it is evidenced again through the frequency plots that this wavelet does not carry enough energy in the frequencies that compose the ECG, which explains its overall poor performance in the classification experiments compared to the others."

Minor remarks:

2.2.1 Beat segmentation - you do not specify this, so I assume you are using R-peak positions taken directly from MIT-BIH annotations. If so, it should be stated.

R: Yes, the reviewer is correct. This information was added to the document at the end of the Beat Segmentation subsection: “The beats were segmented via code using the R-peak annotations provided by the MIT-BIH database.”

2.2.4, page 5 - dimension of 64x64 - please clarify: does this mean a common 64x64 grid was employed for all spectrograms and scalograms? That is, 4096 data values per spectrogram/scalogram?

R: Yes, a standard 64x64 grid was employed for all spectrograms and scalograms, totaling 4096 features, but the values in the interpolation will depend on the raw representations that are extracted from the transforms. 

2.2.4, page 5 - annulment ... possibly "cancellation"?

R: Thanks for the suggestion. We fixed it.

2.2.4, page 6 - spectrum figures are strictly not amplitudes and log (mV^2) is a rather odd choice, as far as units are concerned. Since your values are nevertheless derived from voltage measurements (the ECG traces) I would suggest the use a better suitable logaritmic unit for *amplitude*, namely, dBV = 20 x log_10 (voltage_RMS) - the conversion is rather straightforward if you start from the values of Fourier coefficients

R: Thank you for the suggestion. We had used the log (mV^2) unit in the first place to match the power spectrum in the spectrogram, but dBmV does make more sense and it worked just as well to make our case in the plots. We adjusted the respective figure.

  1. Discussion, page 12 - I find it rather hard to believe that a notch filter at 60 Hz does suppress *any* significant information "useful to differentiate cardiac rythms". Components that are over two orders of magnitude less than the most significant ones may at most affect subtle aspects of the waveform shape, but I'd expect them to be irrelevant to arrythmia.

R: The reviewer is correct. We have even noticed such a thing in the waveform of the ECG, as it barely changes when applying the PLI filter, and this was even commented on by us on page 5 when we discussed the effects of filtering. In light of that, we have crossed out this saying.

  1. Conclusion and Future Works - "CWT is a more robust signal processing technique" - what do you mean by "more robust"? This sounds too generic and might be inappropriate.

R: Sorry about that. We have changed it to “We showed that the CWT works better as a spectral transform for classifying arrhythmia than the STFT, mostly due to its multi-resolution property and flexibility provided by wavelet functions with different frequencies.”

At the end of a rather long list of remarks I do encourage you to carry on, as the basic idea is interesting.

Round 2

Reviewer 2 Report

Good improvement.

Just a few minor points:

Table 3: to help readers follow your discussion, also highest frequency values might be added to Table 3, rather than giving them in the text. In this way, everything is readily visible and comparable.

Page 12, lines 310-312: What you remark is likely related to the fact that the Mexican Hat always reaches lower frequencies than other wavelets. It may be appropriate to link this with later discussion on page 14 (or move it there).

Page 13, line 325: "As far as expainability goes..." sounds redundant, doesn't it?

Page 13, line 336: "which may suggest a higher time dependency in the ECG signals than a frequency dependency" -- This sentence is rather obscure. Maybe you are thinking of time correlation? This might a way to put it in a more understandable way.

Page 15, lines 402-407: "so it is natural to conclude that such filter is not necessary" -- THIS IS NONSENSE -- If a 50/60 Hz disturbance is present, it will affect your estimates (think of the strongly oscillatory Morlet wavelet, for instance). On the other hand, since the signal frequency content tends to be lower, it is appropriate to say that a 50/60 notch filter does not affect your feature extraction accuracy. As to your observation that 60 Hz noise is negligible at any rate in the MIT-BIH waveforms, I'd believe that Holter recoders by which waveforms were acquired already included some form of notch filter. So, saying a notch filter is unnecessary can be misleading/misunderstood.

Author Response

Good improvement.
R: Thanks! The improvements were possible because of the reviewers' contributions.

Just a few minor points:
Table 3: to help readers follow your discussion, also highest frequency values might be added to Table 3, rather than giving them in the text. In this way, everything is readily visible and comparable.

R: Thanks for the suggestion. We have made the proper adjustments to the table to include the scale value of 1. We also changed the column terms from “X scales”  to “S = X” to make it more understandable because we realized it was not so much. We also explained the “S” in the caption. Finally, we crossed out the extra information of the highest frequency values from the text.

Page 12, lines 310-312: What you remark is likely related to the fact that the Mexican Hat always reaches lower frequencies than other wavelets. It may be appropriate to link this with later discussion on page 14 (or move it there).

R: Thanks for the suggestion. We moved it to the paragraph in line 377. The link may be more evident in this new format. 

Page 13, line 325: "As far as expainability goes..." sounds redundant, doesn't it?

R: Thanks for pointing that out. Indeed it is. We have crossed this out.

Page 13, line 336: "which may suggest a higher time dependency in the ECG signals than a frequency dependency" -- This sentence is rather obscure. Maybe you are thinking of time correlation? This might a way to put it in a more understandable way.

R: We agree with the reviewer and we changed the paragraph to make the text clearer. However, the experiment with the lowest average F1-Score is still higher than the other two overlap configurations. Furthermore, the configuration with the lowest time correlation, the 75\% overlap, had the lowest average F1-Score among the other two overlap experiments. It leads us to believe that, despite such a large number of redundant values, the network seems to have a more steady learning procedure in the configuration with time-dependence.

Page 15, lines 402-407: "so it is natural to conclude that such filter is not necessary" -- THIS IS NONSENSE -- If a 50/60 Hz disturbance is present, it will affect your estimates (think of the strongly oscillatory Morlet wavelet, for instance). On the other hand, since the signal frequency content tends to be lower, it is appropriate to say that a 50/60 notch filter does not affect your feature extraction accuracy. As to your observation that 60 Hz noise is negligible at any rate in the MIT-BIH waveforms, I'd believe that Holter recoders by which waveforms were acquired already included some form of notch filter. So, saying a notch filter is unnecessary can be misleading/misunderstood.

R: We agree with the reviewer and changed the phrase to avoid misunderstanding. We changed to "Thus, the 50/60 Hz notch filter does not significantly affect the feature extraction accuracy for arrhythmia classification."
